# Higher Mutation Burden in High Proliferation Compartments of Heterogeneous Melanoma Tumors

**DOI:** 10.3390/ijms22083886

**Published:** 2021-04-09

**Authors:** Tomasz M. Grzywa, Agnieszka A. Koppolu, Wiktor Paskal, Klaudia Klicka, Małgorzata Rydzanicz, Jarosław Wejman, Rafał Płoski, Paweł K. Włodarski

**Affiliations:** 1Center for Preclinical Research, The Department of Methodology, Medical University of Warsaw, 1B Banacha Str., 02-097 Warsaw, Poland; tomasz.grzywa@wum.edu.pl (T.M.G.); wiktor.paskal@wum.edu.pl (W.P.); klaudia.klicka@wum.edu.pl (K.K.); 2Doctoral School, Medical University of Warsaw, 61 Zwirki and Wigury Str., 02-091 Warsaw, Poland; 3Department of Immunology, Medical University of Warsaw, 5 Nielubowicza Str., 02-097 Warsaw, Poland; 4Department of Medical Genetics, Medical University of Warsaw, 3C Pawinskiego Str., 02-106 Warsaw, Poland; agnieszka.koppolu@gmail.com (A.A.K.); malgorzata.rydzanicz@wum.edu.pl (M.R.); rafal.ploski@wum.edu.pl (R.P.); 5Postgraduate School of Molecular Medicine, Medical University of Warsaw, 02-091 Warsaw, Poland; 6Department of Pathology, Medical Center of Postgraduate Education, 00-416 Warsaw, Poland; jarwej@poczta.fm

**Keywords:** intratumor heterogeneity, genetic heterogeneity, melanoma, NGS

## Abstract

Melanoma tumors are the most heterogeneous of all tumor types. Tumor heterogeneity results in difficulties in diagnosis and is a frequent cause of failure in treatment. Novel techniques enable accurate examination of the tumor cells, considering their heterogeneity. The study aimed to determine the somatic variations among high and low proliferating compartments of melanoma tumors. In this study, 12 archival formalin-fixed paraffin-embedded samples of previously untreated primary cutaneous melanoma were stained with Ki-67 antibody. High and low proliferating compartments from four melanoma tumors were dissected using laser-capture microdissection. DNA was isolated and analyzed quantitatively and qualitatively. Libraries for amplicon-based next-generation sequencing (NGS) were prepared using NEBNext Direct Cancer HotSpot Panel. NGS detected 206 variants in 42 genes in melanoma samples. Most of them were located within exons (135, 66%) and were predominantly non-synonymous single nucleotide variants (99, 73.3%). The analysis showed significant differences in mutational profiles between high and low proliferation compartments of melanoma tumors. Moreover, a significantly higher percentage of variants were detected only in high proliferation compartments (39%) compared to low proliferation regions (16%, *p* < 0.05). Our results suggest a significant functional role of genetic heterogeneity in melanoma.

## 1. Introduction

Melanoma is a neoplasm arising from melanocytes, neural crest-derived pigment cells [1]. It is the deadliest type of skin tumor with highly metastatic capabilities and aggressive behavior. Incidence rates continue to increase. Melanoma is the fifth most common skin neoplasm type, with 95,710 new cases in the United States in 2020 [2]. While early-stage melanomas can be cured by surgical excision, advanced metastatic melanoma is associated with short overall survival. Because of the significant development of new therapies, such as checkpoint inhibitors, immunotherapies, and targeted therapies, the mortality rates decrease [2]. Nonetheless, the number of total deaths is estimated for nearly 10,000 cases each year [3].

Melanoma is characterized by high intratumor heterogeneity, i.e., the existence of multiple populations of neoplastic cells with distinct features within one tumor. Intratumor heterogeneity occurs on different levels such as the genome, transcriptome, proteome, and epigenome [4]. The main cause of tumor genetic heterogeneity is the accumulation of mutations by neoplastic cells caused by genomic instability induced by environmental factors, including ultraviolet (UV) radiation or insufficient DNA damage response [4,5]. That genetic and phenotypic variation between cells leads to the selection of new subclones that are resistant to applied therapy [6].

Tumor heterogeneity is one of the causes of therapy failure in patients [4]. However, currently, most of the standard diagnostic procedures underestimate actual clonal tumor composition. Recently, next-generation sequencing (NGS) emerged as a promising tool to determine the profile of mutations and quantify mutational burden [7,8]. Thus, NGS is useful in selecting the appropriate personalized, targeted therapy for patients with melanoma [7,8]. It was demonstrated that the quantity and quality of DNA from archival formalin-fixed and paraffin-embedded (FFPE) tissue is suitable for NGS analysis [9,10,11].

In this study, we used the NGS method to detect genetic variants in a panel of cancer-related genes in compartments of high and low proliferation within primary cutaneous melanoma tumors.

## 2. Results

Samples of 12 previously untreated primary cutaneous melanoma were included in the study (Figure 1). Resected tumors were formalin-fixed and paraffin-embedded according to the standard protocol. The samples were cut on a microtome and stained with hematoxylin and eosin for the pathologist examination according to the seventh edition of AJCC Melanoma Staging and Classification. Subsequent slices were stained with anti-Ki67 antibodies to determine proliferation patterns within a tumor. Melanoma tumors obtained from four patients exhibited significant heterogeneity of Ki-67 staining with both high and low proliferation compartments of tumor cells and were subjected to further studies (Table 1). Tumors resected from eight patients exhibited homogeneously high intensity of Ki-67 staining and were excluded from the study (Appendix A).

Subsequent sections of melanoma tumors were stained with hematoxylin and subjected to laser-capture microdissection (LCM)-aided dissection of two regions of each tumor tissue—exhibiting high proliferation (HP) and low proliferation (LP). The pattern of proliferation was determined based on the density of Ki-67-positive cells revealed by immunohistochemical staining with anti-Ki-67 antibodies (Figure 2). HP and LP compartments were defined as areas of tumor cells with higher (HP) or lower (LP) density of Ki-67-positive cells compared to mean density for the whole tumor slice. Regions of more than 10% Ki-67-positive tumor cells were considered HP (preferably and mainly areas of >50% positive staining were dissected), and compartments of less than 10% Ki-67-positive cells were considered LP.

Next-generation sequencing (NGS) of a commercial panel of cancer-related genes (NEBNext Direct^®^ Cancer HotSpot Panel Table 2) was used to detect mutations (single nucleotide variants (SNVs) and small insertion/deletions). This panel includes oncogenes and tumor suppressor genes with a well-established role in melanoma that regulate hallmarks of cancer, including sustaining proliferative potential and evading growth suppression [12,13,14].

We detected 206 variants in a total number of 42 genes. Most of them (135, 66%) were located within exons. Most variants within exons were non-synonymous single nucleotide variants (SNVs) (99, 73.3%). Synonymous SNVs (25, 18.5%), stopgain SNVs (4, 3.0%), frameshift deletions (3, 2.2%), non-synonymous multi-nucleotide variants (MNVs) (2, 1.4%), frameshift insertions (1, 0.7%), and non-synonymous deletions (1, 0.7%) within the exon were detected in a smaller number. Intronic variants (71, 34%) were predominantly SNVs (55, 77.5%), followed by deletions (11, 15.5%) and insertions (2, 2.8%) as well as untranslated region (UTR) variants (3, 4.2%). A total of 35 (17%) of all variants were classified as UV-signature mutations (C>T, CC>TT [15]).

We found 16 variants in 15 genes in the tumor tissue of Patient 1 (Table 3). Six of them were detected only in HP compartments, two only in LP compartments, and eight variants were detected in both HP and LP regions (shared variants).

We detected 84 variants within 32 genes in the tumor tissue of Patient 2 (Table 4). A total of 39 variants were found only in the HP compartment, 29 were detected only in the LP compartment, and 16 variants were detected in both compartments.

We detected 54 variants within 25 genes in the tumor tissue of Patient 3 (Table 5). A total of 25 variants were found only in the HP compartment, 5 were detected only in the LP compartment, and 24 variants were detected in both compartments.

We detected 52 variants within 25 genes in the tumor tissue of Patient 5 (Table 6). A total of 14 variants were found only in the HP compartment, 3 were detected only in the LP compartment, and 35 variants were detected in both compartments.

Analysis of variants detected in selected compartments revealed higher numbers of variants in HP compared to LP compartments (Figure 3a). Importantly, the percentage of variants detected only in HP compartments was significantly higher than of those detected only in LP (Figure 3b). Numbers and percentages of variants detected only in HP compartments were similar to those detected in both tumor sections (shared variants). Detailed analysis of variant allele frequency (VAF) of each patient revealed significantly higher VAFs in HP compartments than LP (Figure 3c).

Most of the mutations were detected only in HP compartments (Figure 4). Nonetheless, shared variants detected in both HP and LP also constituted a substantial percentage of variants. In contrast, variants detected only in LP compartments were rare.

Variants detected in HP compartments (Figure 5a) were commonly absent in LP (Figure 5b). Nonetheless, shared variants that were detected in both HP and LP constitute a substantial percentage of mutations (Figure 4). Shared variants exhibited either similar VAF (Figure 5c) or VAF was higher in HP compared to LP compartment (Figure 5d).

Our report demonstrates a higher mutational burden in high proliferating compartments of melanoma tumors compared to low proliferating ones.

## 3. Discussion

The development of targeted therapies for melanoma significantly prolonged the overall survival of patients with malignant melanoma [2]. However, a substantial group of patients either do not respond to the therapy or develop acquired resistance and eventually progress [16]. Genetic intratumor heterogeneity is one of the major obstacles to the successful clinical outcome of patients treated with targeted therapies [17].

Melanoma tumors exhibit one of the highest numbers of clones within tumors from all types of neoplasms [18]. Tumor heterogeneity has relevant clinical implications [19], is associated with worsened prognosis [19], and is an important cause of resistance to cancer therapies [20]. In melanoma, tumor heterogeneity is caused by many factors [4], including high mutational load caused by UV radiation [21,22,23]. We found that 17% of variants detected within a panel of cancer-related genes had UV-signature, which was substantially lower compared to over 82% of mutation with UV-signature detected by whole-exome sequencing of melanoma tumors [21]. Nonetheless, in our analysis non-synonymous SNVs constituted the majority of detected variants (73.3%), which is consistent with the result whole-exome sequencing of melanoma tumors [21].

In our study, we reported that high proliferation compartments of melanoma tumors have a higher mutation load in genes with a crucial role in oncogenesis compared to low proliferation regions. So far, it was reported for breast cancer that a higher mutation burden is associated with higher proliferation ability and aggressive clinical features [24]. Likewise, in uveal melanoma higher mutation burden was observed in small tumors that exhibited higher proliferation rates [25]. However, it remained unknown whether tumor cells population with a high proliferation rate have a higher mutational load. Here, we demonstrated that HP compartments of melanoma tumors have higher numbers of mutations as well as higher VAFs compared to LP compartments. For instance, we observed numerous mutations in the *Rb* gene in HP compartments in three patients (5, 7, and 6 variants in Patients 2–4, respectively). On the contrary, Rb variants in LP were less common (2, 1, and 1 in Patients 2–4, respectively). Similarly, mutations in the *TP53* gene were detected mostly in HP compartments (1, 3, 1, and 4 variants in Patients 1–4, respectively). *Rb* and *TP53* are tumor suppressor genes that are critical targets of mutagenesis in melanoma [26,27].

Mutations in platelet-derived growth factor receptor α (*PDGFRA*) were detected only in the HP compartment or were shared in both compartments. *PDGFRA* is a proto-oncogene, and mutations within its gene are detected in about 5% of melanoma tumors [28,29]. Moreover, variants in the *ERBB4* gene, a commonly mutated proto-oncogene in melanoma [30], were detected in higher numbers in HP compartments (1, 5, 2, and 1 variant in Patients 2–4, respectively) than in LP compartments (0, 4, 3, and 0 variant in Patients 2–4, respectively).

Importantly, we observed that low proliferation compartments of melanoma tumors have a different mutational profile compared to high proliferation regions. Within melanoma tumors, the slow-cycling cells exhibit increased resistance to therapies and may trigger a relapse of the disease [31]. Because of the high resistance of melanoma slow-cycling subpopulation to conventional as well as targeted therapies and their ability to reconstitute tumor mass after treatment, there is a need for a better understanding of their mutation profile, which may result in the development of novel, more effective targeted therapies [4,32].

The main limitation of our study was the low number of analyzed patients and a low number of sequenced genes. Moreover, we did not investigate the biological effects of these mutations in melanoma cells. Further studies are required to determine the mutational landscape of distinct regions of melanoma tumors with different features on the whole genome level to provide insights regarding the functional role of genetic heterogeneity of melanoma tumors. Our study suggests that proto-oncogenes and tumor suppressor genes are more commonly mutated in compartments of high proliferating melanoma cells, which may contribute to the accelerated growth.

## 4. Materials and Methods

### 4.1. Patients Tissue

The study was performed on archival formalin-fixed, paraffin-embedded (FFPE) primary cutaneous melanoma tumors originating from 12 patients from the Department of Pathology, Medical Center of Postgraduate Education, Warsaw, Poland. The clinical and histopathological data of patients are presented in Table 1 and Appendix A. The study was conducted in accordance with the Declaration of Helsinki. The study was approved by the Bioethical Committee Medical University of Warsaw (AKBE/301/2019). The detailed protocol of the study is presented in Figure 1.

### 4.2. Hematoxylin and Eosin Staining

Resected skin tumors were formalin-fixed and paraffin-embedded according to the standard protocol in the tissue processor. The FFPE samples were cut on a microtome and stained with hematoxylin and eosin according to the standard diagnostic protocol. Subsequently, they were examined by a board-certified pathologist and reanalyzed by the second pathologist (J.W.) according to the seventh edition of AJCC Melanoma Staging and Classification.

### 4.3. Immunohistochemistry Staining

For immunohistochemical staining, the samples were cut on a microtome (Leica, RM2055 model, Buffalo Grove, IL, USA) on 3 µm slices. In the next step, samples were deparaffinized and rehydrated using xylene and ethanol. To determine the compartments of high and low proliferation patterns, tumors were stained with anti-Ki-67 antibody (NB110-90592, NovusBio, Centennial, CO, USA) at final dilution 1:3200. To confirm the melanocytic origin of the neoplastic cells, samples were stained with anti-MITF antibody (PA538294, Thermofisher Scientific, Waltham, MA, USA) at final dilution 1:300. Immunohistochemistry staining was performed using EnVision™ FLEX DAB+ Substrate Chromogen System (Dako, Agilent, Santa Clara, CA, USA) according to the manufacturer’s protocol.

### 4.4. Preparation for Laser-Capture Microdissection (LCM)

All samples for LCM were cut with a microtome to 10 µm slices (Leica, RM2055) and were mounted on glass slides (SuperFrost Ultra Plus, Menzel Gläser, Thermofisher Scientific [33]) with a drop of DNAse/RNAse-free water. Next, samples were incubated in a fume hood at 56 °C for 1 h to increase adherence to slides. Mounted slices were hematoxylin stained according to the standard protocol in a set of alcohol solutions, xylene, and stain.

### 4.5. Laser-Capture Microdissection

Stained and dehydrated sections of tissues were subjected to LCM-aided dissection, as described before [33,34]. Two regions of each melanoma tissue were selected depending on the intensity of proliferation based on Ki-67 staining (areas with low and high proliferation). Melanomas that exhibited a homogeneous density of Ki-67-positive cells within tumors were excluded from the study. Tumors that had areas with different densities of Ki-67-positive cells (heterogeneous Ki-67 staining) were included for further examination. At least two researchers chose by consensus compartments exhibiting higher density (high proliferation, >20 mitoses in HPF, >10% Ki-67-positive cells, usually >50%) and lower density (low proliferation, <10% Ki-67-positive cells) of Ki-67-positive tumor cells compared to mean density Ki-67-positive cells of whole tumor tissue. Representative scans of high and low proliferation compartments are presented in Figure 2b and Figure 5a. The neoplastic character of dissected tissues was assessed based on pathomorphological features by a board-certified pathologist and confirmed by MITF staining. Subsequently, 5 µm^2^ of each region were marked to dissect with the LCM system (PALM Robo, Zeiss, Oberkochen, Germany). The conditions of LCP (Laser Catapulting Pressure) were as follows: LCP energy—82–92, LCP spot distance—25 μm, magnification—5×, tissue collected in 20 μL of Digestion Buffer (Norgen Biotek FFPE RNA/DNA Purification Plus Kit, Thorold, ON, Canada) in 500 μL sterile PCR-tube cap. Each LCM was preceded by optimization of LCP energy and spot distance to provide an effective dissection of marked areas. Caps were sealed back with tubes, centrifuged briefly, and placed on ice until further steps.

### 4.6. DNA Isolation and Quality Verification

In the next step, samples were digested with proteinase K for 48 h at 37 °C followed by DNA isolation using Norgen Biotek FFPE RNA/DNA Purification Plus Kit according to the manufacturer’s protocol (Cat. 54300). DNA was eluted with 15 µL of ultrapure H_2_O preheated to 90 °C. Quantity of DNA was measured using Qubit Fluorometer and Qubit™ dsDNA HS Assay Kit (ThermoFisher Scientific, Waltham, MA USA). DNA quality was verified using Bioanalyzer 2100 according to the manufacturer protocol.

### 4.7. Library Preparation and Next-Generation Sequencing

A total of 10 ng of isolated DNA was fragmented using Covaris M220 Focused ultrasonicator to obtain 200 bp fragment size. Libraries were prepared according to the protocol of NEBNext Direct^®^ Cancer HotSpot Panel provided by the manufacturer. Accordingly, steps were as follows: denaturation and probe hybridization, 3′ blunting of DNA, dA-tailing, ligation of 3′ adaptor, 5′ blunting of DNA, ligation of 5′UMI adaptor, adaptor cleaving, and PCR amplification. The next-generation sequencing (NGS) was performed using Illumina HiSeq 1500cancer. All steps were conducted according to the manufacturer’s protocol. Reads within 50 cancer-related genes (Table 2) were aligned to the hg38 reference genome sequence. Integrative Genomics Viewer v.2.8 was used to visualize NGS results (IGV, http://software.broadinstitute.org/software/igv/, accessed on 15 January 2021). Pathogenicity of variants was determined with DANN [35].

### 4.8. Statistical Analysis and Data Presentation

Statistical analyses were conducted with GraphPad Prism 8.4.3 (GraphPad Software Inc., San Diego, CA, USA) using the repeated-measures ANOVA with Tukey’s post-hoc test and paired t-test. All values are represented as mean ± SD. A *p-*value of <0.05 was considered statistically significant. Figure 1 and Figure 4 were created with Biorender.com.

## Figures and Tables

**Figure 1 ijms-22-03886-f001:**
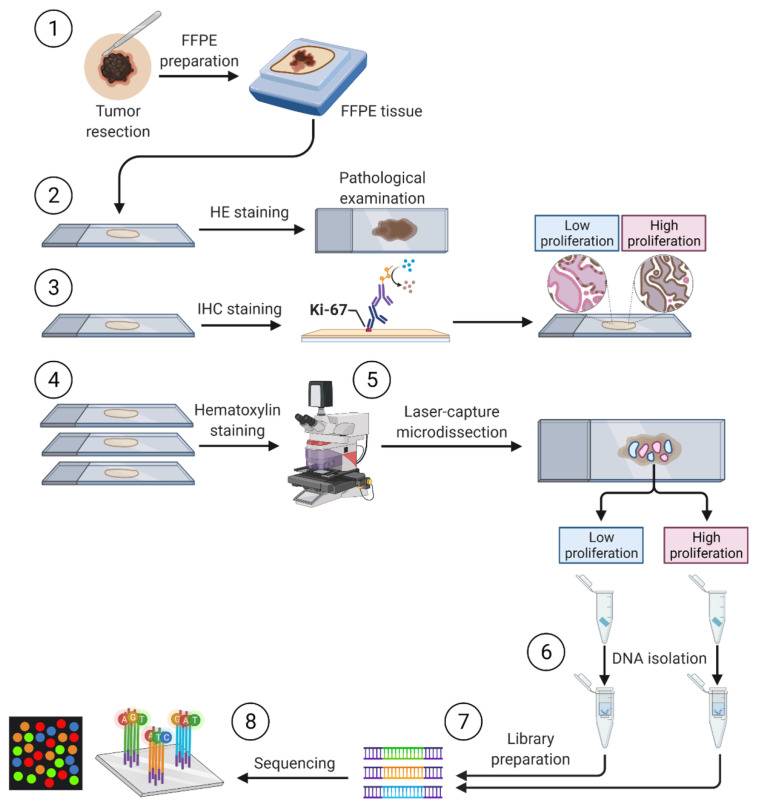
General overview of sample processing. (**1**) Preparation of archival formalin-fixed paraffin-embedded (FFPE) originated from primary cutaneous melanoma tumors. (**2**) Staining with hematoxylin and eosin (HE) for the pathologist examination. (**3**) Immunohistochemistry staining with anti-Ki67 antibodies to determine the compartments of high and low proliferation. (**4**) Hematoxylin staining and laser-capture microdissection. (**5**) The laser-capture microdissection of compartments of high and low proliferation based on Ki-67 staining. (**6**) Digestion with proteinase K and DNA isolation followed by DNA quantity and quality assessment. (**7**) Preparation of libraries for amplicon-based next-generation sequencing of a panel of cancer-related genes. (**8**) Targeted next-generation sequencing.

**Figure 2 ijms-22-03886-f002:**
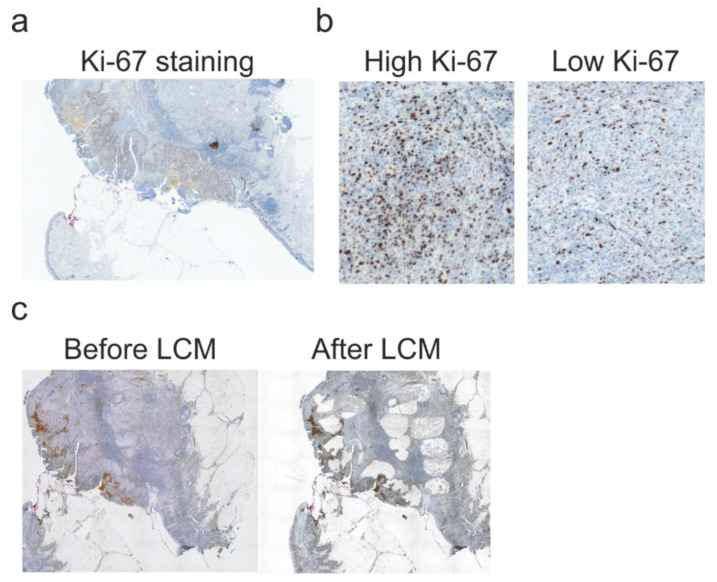
Laser-capture microdissection of chosen compartments of tumor tissue. (**a**). Representative Ki-67 staining of melanoma tissue. Magnification 5×. (**b**). Representative Ki-67 staining of high proliferation (HP) and low proliferation (LP) compartments. HP and LP fragments were defined as a compartment of tumor tissue with a higher or lower density of Ki-67-positive cells compared to the mean density of whole tumor slices. Magnification 20×. (**c**) Scans of the samples before and after laser-capture microdissection (LCM). Magnification 5×.

**Figure 3 ijms-22-03886-f003:**
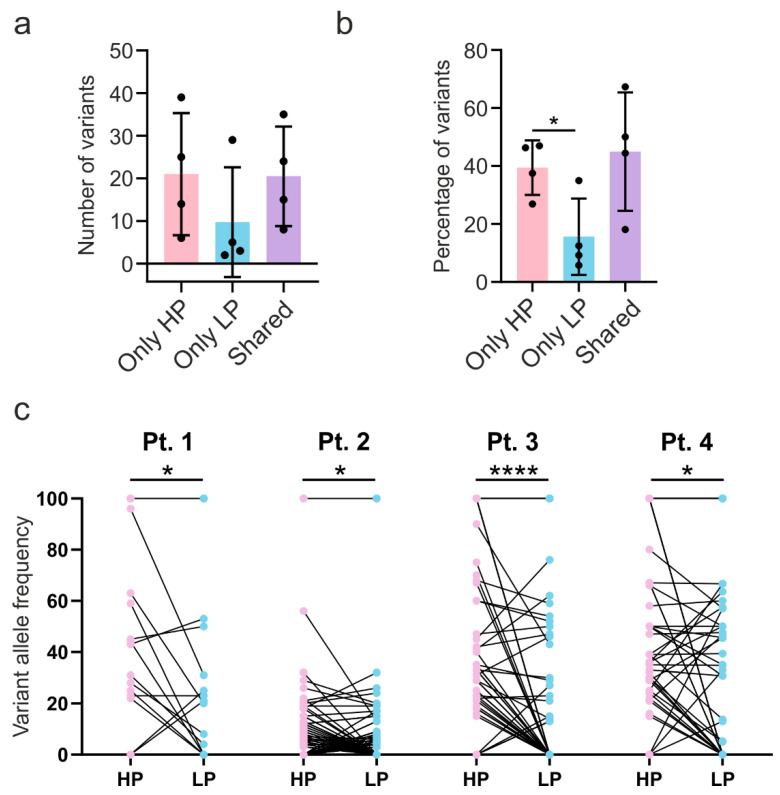
Higher mutation burden in high proliferation compartments. a,b. Number (**a**) and percentage (**b**) of variants detected only in high proliferation (HP) compartment, low proliferation (LP) compartment, and variants detected in both compartments (shared). *p*-value calculated using repeated-measures ANOVA with Tukey’s post-hoc test. (**c**) Variants’ allele frequency detected in high proliferation (HP) compartments compared to low proliferation (LP) compartments in each patient. *p*-value was calculated using paired t-test. * *p* < 0.05, **** *p* < 0.0001

**Figure 4 ijms-22-03886-f004:**
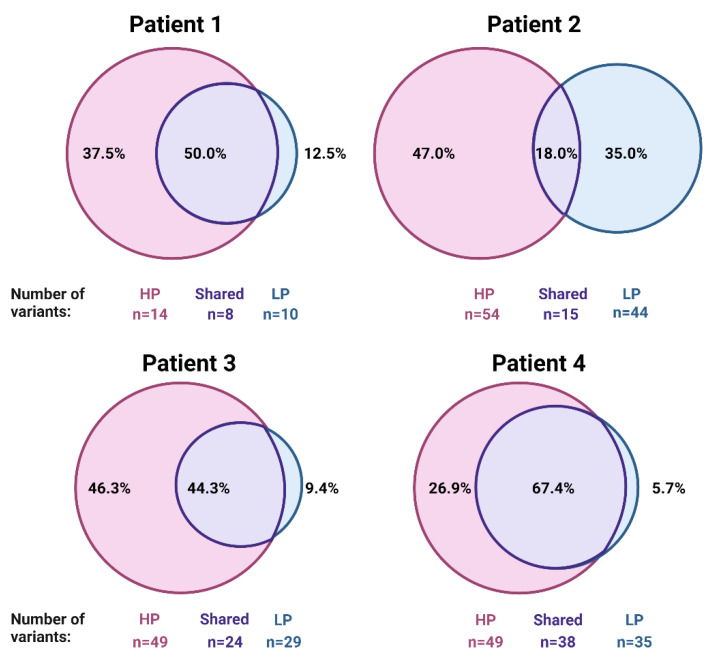
Variants detected in compartments with higher proliferation constitute the majority of variants detected in melanoma tumors. Numbers and percentages of variants detected in high proliferation (HP), low proliferation (LP), and in both compartments (shared).

**Figure 5 ijms-22-03886-f005:**
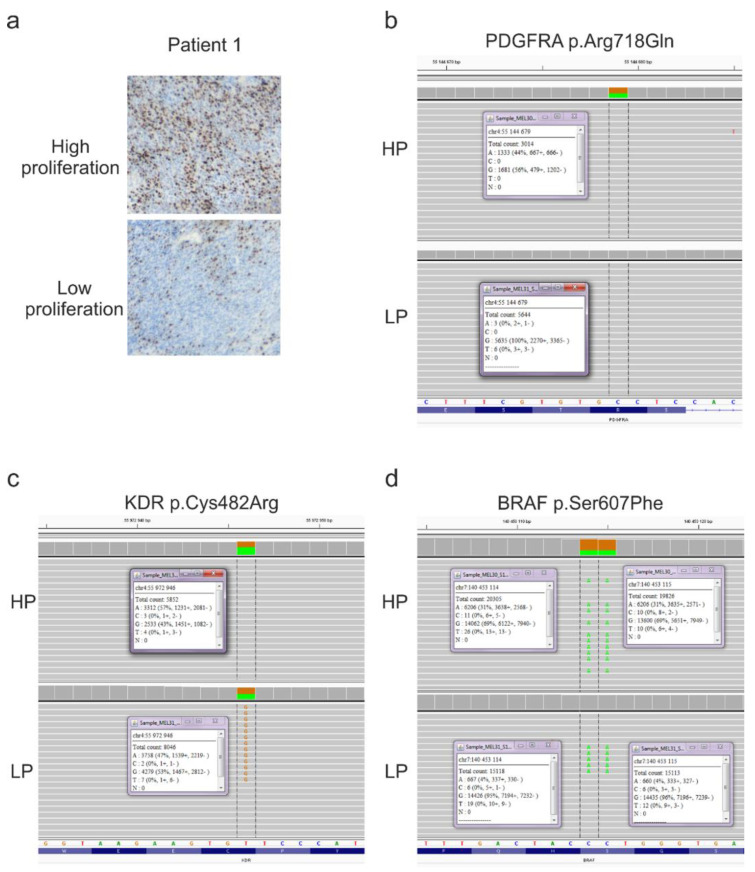
Representative molecular characteristics of patient 1. (**a**) Representative microphotographs of Ki-67 staining of chosen compartments of high proliferation (HP) and low proliferation (LP) of melanoma tissue. Magnification 20×. (**b**) p.Arg718Gln mutation in PDGRFA gene as a representative variant detected only in HP compartment. (**c**) p.Cys482Arg mutation in KDR gene as a representative variant detected in similar variant allele frequency (VAF) in both compartments. (**d**) p.Ser607Phe variant in BRAF gene as a representative variant detected in both compartments but with substantially higher VAF in HP (31%) than LP (4%) compartment.

**Table 1 ijms-22-03886-t001:** Main clinical and histopathological data of the four patients included in the study.

Patient	Patient 1 Mel001	Patient 2 Mel002	Patient 3 Mel010	Patient 4 Mel011
Sex	F	M	F	F
Age	78	81	82	76
Anatomical location	Left cheek	Right cheek	Left eyebrow	Left crus
Histological subtype	Fusocellular NM	Fusocellular NM	SSM	LMM
TNM	pT4b	pT4b	pT1b	pT1a
Clark	V	V	II	II
Breslow	8 mm	7 mm	0.9 mm	0.29 mm
Ulceration	Yes	Yes	Yes	No
Mitotic index	3–6/mm^2^	7/mm^2^	2/mm^2^	1/mm^2^
Lymphoid infiltration	Yes	Yes	Brisk	Brisk
Satellite tumors	In subcutaneous fat tissue	No	No	No
Lymph nodes	n.d.	Clear	n.d.	n.d.
Ki-67	Heterogeneous	Heterogeneous	Heterogeneous	Heterogeneous

HPF—high power field, LMM—lentigo maligna melanoma, MF—mitotic figures, n.d.—no data, NM—nodular melanoma, SSM—superficial spreading melanoma.

**Table 2 ijms-22-03886-t002:** List of cancer-related genes covered by next-generation sequencing panel.

NEBNext Direct^®^ Cancer HotSpot Panel
*ABL1*, *AKT1*, *ALK*, *APC*, *ATM*, *BRAF*, *CDH1*, *CDKN2A*, *CSF1R*, *CTNNB1*, *EGFR*, *ERBB2*, *ERBB4*, *EZH2*, *FBXW7*, *FGFR1*, *FGFR2*, *FGFR3*, *FLT3*, *GNA11*, *GNAQ*, *GNAS*, *HNF1A*, *HRAS*, *IDH1*, *IDH2*, *JAK2*, *JAK3*, *KDR*, *KIT*, *KRAS*, *MET*, *ML1*, *MPL*, *NOTCH1*, *NPM1*, *NRAS*, *PDGFRA*, *KIP3CA*, *PTEN*, *PTPN11*, *RB1*, *RET*, *SMAD4*, *SMARCB1*, *SMO*, *SRC*, *STK11*, *TP53*, *VHL*

**Table 3 ijms-22-03886-t003:** Molecular characteristics of variants detected in tumor tissue of Patient 1.

Gene	Type of Alteration	Variant	Amino Acid Change	Pathogenicity	VAF HP	VAF LP
Shared variants						
*KDR*	Intronic SNV	c.*27T>C	-	-	100%	100%
*KDR*	Non-synonymous MNV	c.3433GG>AA	p.Gly1145Lys	0.9889	23%	23%
*KDR*	Non-synonymous SNV	c.2699A>G	p.Asn900Ser	0.9998	63%	20%
*KDR*	Non-synonymous SNV	c.1444T>C	p.Cys482Arg	0.9998	43%	53%
*KDR*	Non-synonymous SNV	c.1416A>T	p.Gln472His	0.0797	45%	50%
*NPM1*	Intronic deletion	c.*165delT	-	-	100%	100%
*FLT3*	Intronic SNV	c.1310T>C	-	0.0232	100%	100%
*TP53*	Non-synonymous SNV	c.215C>G	p.Pro72Arg	0.3636	96%	31%
High proliferation only						
*BRAF*	Non-synonymous MNV	c.1820CC>TT	p.Ser607Phe	0.9898	31%	4%
*CDKN2A*	Non-synonymous SNV	c.341C>T	p.Pro114Leu	0.9999	59%	n.d.
*ERBB4*	Non-synonymous SNV	c.518C>T	p.Ser173Phe	0.9899	28%	8%
*KDR*	Non-synonymous SNV	c.2836C>T	p.Arg946Cys	0.9779	25%	n.d.
*PDGFRA*	Non-synonymous SNV	c.2153G>A	p.Arg718Gln	0.9999	44%	n.d.
*SMAD4*	Intronic SNV	c.1882+4811C>A	-	-	22%	n.d.
Low proliferation only						
*BRAF*	Non-synonymous SNV	c.1406G>A	p.Gly469Glu	0.9999	n.d.	45%
*STK11*	Non-synonymous SNV	c.968C>A	p.Pro323Gln	0.9989	n.d.	21%

HP—high proliferation, LP—low proliferation, n.d.—not detected, MNV—multi-nucleotide variant, SNV—single nucleotide variant, VAF—variant allele frequency.

**Table 4 ijms-22-03886-t004:** Molecular characteristics of variants detected in tumor tissue of Patient 2.

Gene	Type of Alteration	Variant	Amino Acid Change	Pathogenicity	VAF HP	VAF LP
Shared variants						
*EGFR*	Non-synonymous SNV	c.298C>T	p.Pro100Ser	0.9998	12%	15%
*EZH2*	Non-synonymous SNV	c.1922A>T	p.Tyr641Phe	0.9874	15%	17%
*PTEN*	Non-synonymous SNV	c.804C>A	p.Asp268Glu	0.5869	18%	32%
*PTEN*	Non-synonymous SNV	c.810G>T	p.Met270Ile	0.9921	21%	26%
*HRAS*	Non-synonymous SNV	c.145G>A	p.Glu49Lys	0.9989	26%	19%
*ATM*	Non-synonymous SNV	c.8094A>T	p.Leu2698Phe	0.9985	5%	7%
*HNF1A*	Frame shifting insertion	c.864_865insC	p.Gly288_Pro289		4%	4%
*FLT3*	Intronic SNV	c.1310-3T>C	-	0.5447	100%	100%
*RB1*	Frame shifting deletion	c.2107delA	p.Ile703		6%	5%
*TP53*	Non-synonymous SNV	c.215C>G	p.Pro72Arg	0.5704	100%	100%
*ERBB4*	Intronic insertion	c.884_885insT	-		20%	24%
*ERBB4*	Intronic deletion	c.884delT	-		19%	19%
*ERBB4*	Non-synonymous SNV	c.490C>A	p.Gln164Lys	0.9055	7%	6%
*FAIM*	Intronic SNV	c.-60T>C	-	0.9839	29%	20%
*PIK3CA*	Non-synonymous SNV	c.881A>T	p.Tyr294Phe	0.9897	56%	13%
*APC*	Non-synonymous SNV	c.3355C>T	p.His1119Tyr	0.999	12%	15%
High proliferation only						
*APC*	Non-synonymous SNV	c.2876C>T	p.Ser959Phe	0.9857	22%	n.d.
*APC*	Non-synonymous SNV	c.3479C>A	p.Thr1160Lys	0.9702	6%	n.d.
*APC*	Non-synonymous SNV	c.3485A>T	p.Tyr1162Phe	0.9839	12%	n.d.
*APC*	Non-synonymous SNV	c.4749G>C	p.Met1583Ile	0.9963	10%	n.d.
*BRAF*	Non-synonymous SNV	c.1768G>T	p.Val590Leu	0.9982	8%	n.d.
*CSF1R*	Non-synonymous SNV	c.985C>A	p.Pro329Thr	0.9971	9%	n.d.
*ERBB4*	Stopgain SNV	c.2798T>A	p.Leu933*	0.9944	12%	n.d.
*ERBB4*	Non-synonymous SNV	c.1829C>A	p.Pro610Gln	0.7498	7%	n.d.
*FBXW7*	Non-synonymous SNV	c.1186G>T	p.Val396Phe	0.9956	7%	n.d.
*FGFR3*	Non-synonymous SNV	c.1918C>T	p.Arg640Trp	0.9969	6%	n.d.
*GNA11*	Non-synonymous SNV	c.629G>A	p.Arg210Gln	0.9984	10%	n.d.
*HNF1A*	Non-synonymous SNV	c.955G>A	p.Gly319Ser	0.9966	5%	n.d.
*HRAS*	Non-synonymous SNV	c.121C>T	p.Arg41Trp	0.9989	3%	n.d.
*KDR*	Stopgain SNV	c.2959G>T	p.Glu987*	0.9971	7%	n.d.
*KIT*	Non-synonymous SNV	c.1463C>T	p.Thr488Met	0.9992	13%	n.d.
*KIT*	Non-synonymous SNV	c.2056C>T	p.Arg686Cys	0.9991	8%	n.d.
*MET*	Non-synonymous SNV	c.439C>T	p.Pro147Ser	0.9237	32%	n.d.
*MET*	Non-synonymous SNV	c.681G>T	p.Met227Ile	0.9639	5%	n.d.
*MET*	Intronic SNV	c.1201T>C	-	0.9279	12%	n.d.
*MET*	Non-synonymous SNV	c.3650C>G	p.Thr1217Arg	0.9949	4%	n.d.
*NOTCH1*	Non-synonymous SNV	c.7436C>A	p.Ala2479Asp	0.9269	18%	n.d.
*NOTCH1*	Non-synonymous SNV	c.6972C>A	p.Asn2324Lys	0.9753	19%	n.d.
*NOTCH1*	Non-synonymous SNV	c.6229G>A	p.Ala2077Thr	0.9991	3%	n.d.
*NOTCH1*	Non-synonymous SNV	c.5069C>T	p.Ser1690Leu	0.9968	8%	n.d.
*NOTCH1*	Non-synonymous SNV	c.7602G>T	p.Glu2534Asp	0.9916	6%	n.d.
*PDGFRA*	Non-synonymous SNV	c.2470G>A	p.Val824Ile	0.9991	5%	n.d.
*PIK3CA*	Non-synonymous SNV	c.2152A>T	p.Ile718Phe	0.721	15%	n.d.
*PIK3CA*	Non-synonymous SNV	c.995G>T	p.Ser332Ile	0.9736	6%	n.d.
*RB1*	Intronic SNV	c.1696G>T	-	0.9956	5%	n.d.
*RB1*	Non-synonymous SNV	c.2002C>T	p.Arg668Cys	0.9951	4%	n.d.
*RB1*	Non-synonymous SNV	c.2032C>A	p.His678Asn	0.9911	4%	n.d.
*RB1*	Non-synonymous SNV	c.2242G>A	p.Glu748Lys	0.9956	11%	n.d.
*SMAD4*	Non-synonymous SNV	c.1486C>T	p.Arg496Cys	0.9993	15%	n.d.
*SMO*	Non-synonymous SNV	c.1198C>T	p.Arg400Cys	0.9994	4%	n.d.
*SMO*	Non-synonymous SNV	c.595C>T	p.Arg199Trp	0.9992	4%	n.d.
*STK11*	Non-synonymous SNV	c.589G>T	p.Val197Leu	0.9975	9%	n.d.
*STK11*	Intronic SNV	c.598C>A	-	0.9148	5%	n.d.
*TP53*	Non-synonymous SNV	c.839G>A	p.Arg280Lys	0.9977	13%	n.d.
*TP53*	Intronic SNV	c.673G>T	-	0.9944	5%	n.d.
Low proliferation only						
*ERBB4*	Non-synonymous SNV	c.1003G>T	p.Asp335Tyr	0.9837	n.d.	4%
*VHL*	Non-synonymous SNV	c.4C>A	p.Pro2Thr	0.9225	n.d.	9%
*KIT*	Non-synonymous SNV	c.311G>T	p.Ser104Ile	0.9838	n.d.	4%
*KDR*	Non-synonymous SNV	c.1473C>A	p.Phe491Leu	0.5998	n.d.	6%
*APC*	Non-synonymous SNV	c.3192G>T	p.Glu1064Asp	0.9648	n.d.	9%
*APC*	Non-synonymous SNV	c.4749G>T	p.Met1583Ile	0.9961	n.d.	9%
*EGFR*	Non-synonymous SNV	c.1804G>A	p.Glu602Lys	0.9989	n.d.	4%
*EGFR*	Non-synonymous SNV	c.2492G>A	p.Arg831His	0.9965	n.d.	4%
*EGFR*	Non-synonymous SNV	c.2495G>A	p.Arg832His	0.9972	n.d.	5%
*MET*	Stopgain SNV	c.760G>T	p.Glu254*	0.9963	n.d.	13%
*MET*	Non-synonymous SNV	c.1147G>T	p.Val383Leu	0.9872	n.d.	8%
*SMO*	Non-synonymous SNV	c.1246G>T	p.Gly416Cys	0.9962	n.d.	4%
*FGFR1*	Non-synonymous SNV	c.936G>T	p.Lys312Asn	0.9988	n.d.	7%
*NOTCH1*	Non-synonymous SNV	c.6733G>A	p.Gly2245Arg	0.9296	n.d.	3%
*NOTCH1*	Non-synonymous SNV	c.4987C>T	p.Arg1663Trp	0.9989	n.d.	6%
*NOTCH1*	Non-synonymous SNV	c.4793G>T	p.Arg1598Leu	0.9909	n.d.	4%
*PTEN*	Non-synonymous SNV	c.25G>T	p.Val9Phe	0.9949	n.d.	17%
*FGFR2*	Non-synonymous SNV	c.1273C>T	p.Arg425Trp	0.9992	n.d.	4%
*ATM*	Non-synonymous SNV	c.3853G>T	p.Asp1285Tyr	0.99	n.d.	4%
*PTPN11*	Non-synonymous SNV	c.1462A>T	p.Ile488Phe	0.9901	n.d.	13%
*HNF1A*	Non-synonymous SNV	c.528G>T	p.Gln176His	0.9952	n.d.	7%
*RB1*	Stopgain SNV	c.585G>A	p.Trp195*	0.9946	n.d.	9%
*AKT1*	Non-synonymous SNV	c.73C>T	p.Arg25Cys	0.9992	n.d.	4%
*TP53*	Non-synonymous SNV	c.845G>A	p.Arg282Gln	0.9994	n.d.	6%
*STK11*	Non-synonymous SNV	c.196G>T	p.Val66Leu	0.9916	n.d.	5%
*STK11*	Non-synonymous SNV	c.758A>G	p.Tyr253Cys	0.9984	n.d.	3%
*GNAS*	Non-synonymous SNV	c.654C>A	p.Asn218Lys	0.9924	n.d.	4%
*GNAS*	Non-synonymous SNV	c.674G>T	p.Gly225Val	0.9977	n.d.	5%
*GNAS*	Non-synonymous SNV	c.718G>A	p.Asp240Asn	0.9966	n.d.	3%

HP—high proliferation, LP—low proliferation, n.d.—not detected, SNV—single nucleotide variant, VAF—variant allele frequency.

**Table 5 ijms-22-03886-t005:** Molecular characteristics of variants detected in tumor tissue of Patient 3.

Gene	Type of Alteration	Variant	Amino Acid Change	Pathogenicity	VAF HP	VAF LP
Shared mutations						
*NRAS*	Non-synonymous SNV	c.38G>T	p.Gly13Val	0.9973	35%	29%
*HRAS*	Intronic SNV	c.111+15G>A	-	0.8192	47%	50%
*KRAS*	Non-synonymous SNV	c.283C>A	p.His95Asn	0.9035	22%	23%
*FLT3*	Intronic SNV	c.1310-3T>C	-	0.5447	45%	76%
*RB1*	Intronic SNV	c.137+86T>C	-	0.5128	100%	100%
*VHL*	Synonymous SNV	c.216C>A	p.Ser72=	0.9595	33%	30%
*MLH1*	Intronic SNV	c.1039-8T>A	-	0.7774	31%	46%
*TP53*	Non-synonymous SNV	c.215G>C	p.Arg72Pro	0.5704	100%	100%
*PIK3CA*	Intronic SNV	g.2756T>G	-	0.5439	100%	100%
*PDGFRA*	Synonymous SNV	c.1701A>G	p.Pro592=	0.3955	100%	100%
*ERBB4*	Intronic SNV	c.742-37T>A	-	0.6607	23%	21%
*PIK3CA*	Non-synonymous SNV	c.1173A>G	p.Ile391Met	0.9296	41%	62%
*PIK3CA*	Intronic SNV	c.2016-27A>T	-	0.1628	60%	54%
*FGFR3*	Synonymous SNV	c.1956G>A	p.Thr652=	0.7994	100%	100%
*FGFR3*	Intronic SNV	c.1959+22G>A	-	0.9115	68%	59%
*KIT*	Non-synonymous SNV	c.1621A>G	p.Met541Val	0.4908	60%	52%
*KDR*	3′ UTR Variant	c.*27=	-	0.5891	100%	100%
*APC*	Synonymous SNV	c.4425G>A	p.Thr1475=	0.7715	42%	47%
*MET*	Non-synonymous SNV	c.3029C>T	p.Thr1010Ile	0.9992	90%	43%
*SMO*	Intronic SNV	c.538-26C>A	-	0.6804	100%	100%
*SMO*	Intronic SNV	c.747+24G>C	-	0.5792	100%	100%
*SMO*	Synonymous SNV	c.1164G>C	p.Gly258=	0.7710	100%	100%
*EZH2*	Intronic SNV	c.1852-21A>G	-	0.5855	100%	100%
*NOTCH1*	Non-synonymous deletion	c.5015delG	pArg1431Pro	0.9973	29%	14%
High proliferation only						
*ATM*	Frame shifting deletion	c.911delA	pGlu304Gly	0.2396	33%	n.d.
*FLT3*	Intronic SNV	c.1253-6G>A	-	0.4703	35%	n.d.
*RB1*	Synonymous SNV	c.1071A>T	pPro357=		19%	n.d.
*RB1*	Intronic deletion	c.1389+8delA	-	0.6691	29%	n.d.
*RB1*	Intronic SNV	c.1389+16T>A	-		40%	n.d.
*RB1*	Intronic deletion	c.2106+54_2106+56delTTC	-	0.8097	100%	n.d.
*RB1*	Intronic SNV	c.2211+32T>A	-	0.6422	18%	n.d.
*RB1*	Intronic SNV	c.2325+18T>C	-	0.6683	25%	n.d.
*SMAD4*	Intronic SNV	c.956-18C>T	-	0.2969	29%	n.d.
*SMAD4*	Intronic SNV	c.1309-35A>T	-	0.6793	18%	n.d.
*GNA11*	Synonymous SNV	c.771C>T	p.Thr257=	0.9983	100%	n.d.
*ERBB4*	Non-synonymous SNV	c.242G>A	p.Arg81Gln	0.7493	25%	n.d.
*SRC*	Synonymous SNV	c.1508C>T	p.Arg503=	0.9584	23%	n.d.
*MLH1*	Intronic SNV	c.1409+2T>A	-		16%	n.d.
*PIK3CA*	Frame shifting deletion	c.57delA	p.Arg19	0.9528	15%	n.d.
*PIK3CA*	Non-synonymous SNV	c.990T>A	p.Ile330Lys	0.4458	21%	n.d.
*PIK3CA*	Intronic SNV	c.1404+19T>A	-		70%	n.d.
*FGFR3*	Intronic deletion	c.1076-44delG	-	0.5625	35%	n.d.
*KIT*	Intronic SNV	c.2484+78T>C	-	0.8905	75%	n.d.
*APC*	Non-synonymous SNV	c.3112A>T	p.Ile1038Leu	0.9055	33%	n.d.
*APC*	Non-synonymous SNV	c.7820 G>T	p.Ser2607Ile	0.8807	100%	n.d.
*FGFR1*	Synonymous SNV	c.2130C>T	p.Phe710=	0.8019	67%	n.d.
*CDKN2A*	Non-synonymous SNV	c.371G>T	p.Arg73Leu	0.5877	18%	n.d.
*NOTCH1*	Intronic SNV	c.5018+79 G>T	-	0.5021	100%	n.d.
*NOTCH1*	Intronic SNV	c.5018+55C>T	-	0.2396	22%	n.d.
Low proliferation only						
*ATM*	Synonymous SNV	c.8015_c.8018delACC	p.Asp2672=		n.d.	13%
*ATM*	Non-synonymous SNV	c.8021insT	p.Gly2675Trp		n.d.	13%
*ERBB4*	Intronic SNV	c.1717-10G>A	-	0.7227	n.d.	13%
*ERBB4*	Intronic SNV	c.1717-16G>A	-	0.3892	n.d.	15%
*FGFR4*	Intronic SNV	c.728-12C>T	-	0.7287	n.d.	27%

HP—high proliferation, LP—low proliferation, n.d.—not detected, SNV—single nucleotide variant, VAF—variant allele frequency.

**Table 6 ijms-22-03886-t006:** Molecular characteristics of variants detected in tumor tissue of Patient 4.

Gene	Type of Alteration	Variant	Amino Acid Change	Pathogenicity	VAF HP	VAF LP
Shared mutations						
*PDGFRA*	Synonymous SNV	c.1701A>G	p.Pro567=	0.3955	100%	100%
*TP53*	Non-synonymous SNV	c.98C>G	p.Pro72Arg	0.5704	100%	100%
*KDR*	3′ UTR Variant	c.*27=	-	0.5891	100%	100%
*SMO*	Intronic SNV	c.747+24G>C	-	0.5792	100%	100%
*NOTCH1*	Synonymous SNV	c.6555C>T	p.Asp1944=	0.8372	66%	50%
*FGFR3*	Synonymous SNV	c.1956G>A	p.Thr651=	0.7994	100%	100%
*NOTCH1*	Intronic SNV	c.5018+55C>T	-	0.5021	58%	60%
*FLT3*	Intronic SNV	c.1310-3T>C	-	0.5447	100%	100%
*SMO*	Intronic SNV	c.538-26C>T	-	0.6804	100%	100%
*SMO*	Synonymous SNV	c.1164G>C	p.Gly258=	0.7710	100%	100%
*RB1*	Intronic SNV	c.137+86T>C		0.5128	100%	100%
*RET*	Non-synonymous SNV	c.2071G>A	p.Gly691Ser	0.8595	39%	14%
*PDGFRA*	Synonymous SNV	c.2472C>T	p.Val824=	0.7581	39%	39%
*PI3KCA*	Intronic SNV	g.2756T>G	-	0.5439	50%	46%
*RET*	Synonymous SNV	c.2712C>G	p.Ser650=	0.7414	25%	5%
*HRAS*	Synonymous SNV	c.81T>C	p.His27=	0.8165	50%	31%
*APC*	Synonymous SNV	c.4425G>A	p.Thr1475=	0.7715	39%	58%
*PI3KCA*	Intronic SNV	g.2645G>A	-	0.8456	35%	35%
*EZH2*	Intronic SNV	c.1852-21T>C	-	0.5855	36%	50%
*RET*	Synonymous SNV	c.2307G>T	p.Leu769=	0.7308	30%	5%
*HRAS*	Intronic deletion	c.-53-35_ -53-40delCCCAGC	-		67%	67%
*NOTCH1*	Synonymous SNV	c.5094C>T	p.Asp1457=	0.9222	47%	48%
*KDR*	Non-synonymous SNV	c.1416A>T	p.Gln472His	0.7338	33%	31%
*EGFR*	Synonymous SNV	c.2361G>A	p.Gln787=	0.9439	50%	35%
*FGFR4*	Intronic SNV	c.728-35G>A	-	0.6556	100%	100%
*HNF1A*	Synonymous SNV	c.864G>C	p.Gly288=	0.8951	36%	64%
*HNF1A*	Intronic SNV	c.955+94T>G	-	0.3713	50%	50%
*PTEN*	Intronic SNV	c.1026+32T>G	-	0.3730	100%	100%
*CLEC2D*	3′ UTR Variant	c.*1413=	-	0.8819	29%	45%
*PDGFRA*	Intronic SNV	c.2440-50T>TA	-	-	80%	57%
*TP53*	Intronic SNV	c.672+62A>G	-	0.6480	100%	100%
*KDR*	Intronic SNV	c.798+54C>T	-	0.6404	100%	100%
*ATM*	Non-synonymous SNV	c.7391T>A	p.Leu2463Phe	0.9887	21%	47%
*HNF1A*	Intronic SNV	c.527-23C>T	-	0.6046	25%	67%
*NOTCH3*	Synonymous SNV	c.4563A>T	p.Pro1469=	0.6682	100%	100%
Only high proliferation						
*ALK*	Synonymous SNV	c.27C>G	p.Leu9=	0.5627	100%	n.d.
*ERBB4*	Intronic deletion	c.884-7_884-8delAA	-	-	16%	n.d.
*KDR*	Intronic SNV	c.3405-92A>C	-	0.5467	67%	n.d.
*KDR*	Intronic insertion	c.2615-37insC	-	-	100%	n.d.
*KIT*	Intronic SNV	c.2484+78T>C	-	0.5625	100%	n.d.
*NPM1*	Intronic deletion	c.847-17delT	-	-	32%	n.d.
*PIK3CA*	Non-synonymous SNV	c.989T>A	p.Ile330Lys	0.9528	22%	n.d.
*PTEN*	Intronic deletion	c.802-17delT	-	-	24%	n.d.
*PTEN*	Non-synonymous SNV	c.810G>T	p.Gln97His	0.9973	29%	n.d.
*RB1*	Intronic SNV	c.2211+32T>A	-	0.8097	15%	n.d.
*RB1*	Intronic SNV	c.2212-15A>C	-	0.2710	21%	n.d.
*RET*	Synonymous SNV	c.2307G>T	p.Leu769=	0.7308	30%	n.d.
*TP53*	Intronic SNV	c.-28-13A>G	-	0.7886	67%	n.d.
*TP53*	Intronic deletion	c.96+48_97-58del CCCCAGCCCTCCAGGT	-	-	100%	n.d.
Low proliferation						
*PTEN*	Non-synonymous SNV	c.983C>A	p.Ala327Glu		n.d.	57%
*PIK3CA*	Intronic deletion	c.2667-13_2667-14delTA	-	-	n.d.	13%
*CDKN2A*	Intronic SNV	c.457+18C>T	-	-	n.d.	33%

HP—high proliferation, LP—low proliferation, n.d.—not detected, SNV—single nucleotide variant, VAF—variant allele frequency.

## Data Availability

Data is contained within the article or Appendix A.

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
