# Peer review of "Higher Mutation Burden in High Proliferation Compartments of Heterogeneous Melanoma Tumors"

_ijms, 2021, doi:10.3390/ijms22083886_

Round 1
Reviewer 1 Report
Dear Author,
The study is well observed and designed well. You have showed significant differences in mutational profiles between high and low proliferation compartments of melanoma tumors, suggest that significant role of genetic heterogeneity in melanoma. I have few concern about the manuscript.
The minor comments are:
- Fig 2 a, c images are not clear, if they can provide higher resolution will be better and the author have to make arrow mark where they are stained.
- If author can provide the quantification of Fig 2b? and the figure legends not explained about the samples in detail.
- Fig 5 a, Image is not clear.
Please do the changes.
Author Response
"Please see the attachment."

Reviewer 2 Report
Genetic intratumor heterogeneity is one of the major obstacles to accurately determine the line of treatment/predict outcome in melanoma treatments. In this study, the authors aimed to compare somatic variations among high and low proliferating compartments of melanoma tumors. Despite the intriguing premise, this study falls sort in major areas highlighted below. See my detailed comments in the attachment.
- The criteria for determining low Vs high p[roliferating areas are not clear.
- If Ki67 was used as a marker to determine low and high proliferating areas, it is not clear as to how Ki67 homogeneity could help identify low proliferating areas.
- Materials and methods could be improved.
- Figure legends are brief and do not provide enough information
- Discussion is brief and vague and missed key areas a) discussion of the results, b) explaining the implications of the findings, c) limitation of study with one glaring issue of very low n (4).
This being said I applaud the authors for the gorgeous and simple schematic. It is a very effective representation of the workflow.

Author Response
"Please see the attachment."

Round 2
Reviewer 2 Report
I sincerely thank the authors for the rebuttal and rigorous revision.
I have a couple of comments :
-HP and LP compartments were defined as areas of tumor cells with higher (HP) or lower (LP) density of Ki-67-positive cells compared to mean density for whole tumor slice.
- Ki67 staining: I thank the authors for adding the explanation/criteria for defining high/low Ki-67 compartments. However, this is still qualitative. Since the entire paper stands on observations made on this classification, the authors should provide a clear cutoff for HP vs LP. for example "a standard IHC scoring such as 75-90% positive cells/ area or such and such density was considered a high compartment and density below such and such or less than 5% positive cells/tumor slice was considered LP......"
2. limitations. While the sample size is the most obvious, the authors should add another limitation beyond that.
3. Minor English edits are required.
